# Salivary Osteocalcin as Potential Diagnostic Marker of Periodontal Bone Destruction among Smokers

**DOI:** 10.3390/biom10030380

**Published:** 2020-03-01

**Authors:** Betsy Joseph, Mukhatar Ahmed Javali, Mohasin Abdul Khader, Saad M. AlQahtani, Amanullah Mohammed

**Affiliations:** 1Department of Periodontics and Community Dental Sciences, College of Dentistry, King Khalid University, Abha 61421, Saudi Arabia; 2Department of Clinical Biochemistry, College of Medicine, King Khalid University, Abha 61421, Saudi Arabia; amanullahmohammed@yahoo.com

**Keywords:** salivary, osteocalcin (OC), osteonectin (ON), C-terminal telopeptide region of type I collagen (CTX), alveolar bone loss (BL), bleeding on probing (BOP), probing pocket depth (PPD), periodontitis, smokers

## Abstract

The objective of the study was to assess the levels and diagnostic accuracy of salivary osteocalcin (OC), osteonectin (ON), and deoxypyridinoline-containing degradation fragment of the C-terminal telopeptide region of type I collagen (CTX) in adult smokers with periodontal bone destruction. Towards this, ninety systemically healthy patients (groups I: healthy, II: periodontitis with non-smokers, and III: periodontitis with current smokers) were included in the study. The results showed a positive correlation (weak to moderate) was observed for OC, ON, and CTX with probing pocket depth (PPD; r = 0.40, 0.32, and 0.36) and alveolar bone loss (BL; r = 0.58, 0.38, and 0.51) (*p* < 0.01). Smoker periodontitis was best discriminated from healthy controls using 15.25 ng/mL of OC (AUC: 0.870; 95% CI: 0.757–0.943; YI (Youden Index): 0.693; *p* < 0.0001). However, with a cut-off of BL at 33.33%, 19.24 ng/mL of salivary OC gave the best discrimination (AUC: 0.809; 95% CI: 0.686–0.900; Se: 80.0%; Sp: 73.47%, and YI: 0.534). A 16.45 ng/mL amount of OC gave excellent discrimination (AUC: 0.811; 95% CI: 0.688–0.901; Se: 92.31%; Sp: 65.22%, and YI: 0.575) among healthy and smoker periodontitis when PD at 6mm was considered as cut-off. Conclusion: The best discrimination between healthy controls and smoker periodontitis was obtained at 15.25 ng/mL of salivary OC.

## 1. Introduction

In the recent past, various components of saliva have been investigated as biomarkers for screening cases of periodontitis [1]. Periodontitis is an infectious disease of the periodontium in which a cascade of immune-inflammatory events occurs that leads to the degradation of connective tissue and alveolar bone. Proinflammatory mediators such as interleukins, prostaglandin, and C-reactive protein (CRP) play a major role in the etiopathogenesis. Recent evidence shows that during periodontitis, salivary metabolites in gingival tissue play a role in the endothelial damage and impaired vasodilatation mediated by endothelin (ET-1) and nitric oxide (NO) as dysfunction of ET-1 inhibits NO synthase [2]. The progression of oral dysbiosis is also linked to factors such as antioxidants and vitamin C [3] that involves complex processes of signaling between the immune system and bone. Various cells and biochemical molecules including osteoblasts, receptor activator of nuclear factor kappa B ligand, osteoprotegerin, osteoclasts, bone morphogenetic proteins, resorption products of organic skeletal matrix, and inorganic skeletal matrix markers are involved [4]. During periodontal disease, osteoclasts have been found to degrade different components of bone resulting in degradation of type I collagen by enzymes such as matrix metalloproteinases (MMPs). Thus, cross-linked telopeptides enter into the circulation as stable fragments such as C-terminal type I collagen telopeptide (β-CTX) [5]. Management of periodontal disease includes various strategies such as regenerative techniques using bone grafts, stem cells for bone remodeling/repairing [6,7], and approaches using nanotechnologies [8].

Although several studies [9,10,11] have investigated various biomarkers such as osteocalcin (OC), osteonectin (ON), and deoxypyridinoline-containing degradation fragment of the C-terminal telopeptide region of type I collagen (CTX), there are still no conclusive results regarding the most appropriate biomarker of bone turnover in adults with periodontitis. Our recent study reported the varying diagnostic accuracy of these biomarkers for the screening of alveolar bone loss due to periodontitis [12]. While some studies considered increased levels of CTX in gingival crevicular fluid to be of potential diagnostic value in identifying periodontal disease with reasonable accuracy, few others have found CTX to be below the level of detection in most subjects [13,14,15].

OC is one of the most abundant proteins in human bone, and the osteoblasts synthesize it. Not only is OC associated with inhibition of bone mineralization but also various aspects of cognition, insulin sensitivity, energy metabolism, and reproduction [16]. It is a good indicator of alveolar bone destruction in patients with conditions such as periodontal disease, osteoporosis, and in post-menopausal women [17,18,19]. Another biomarker implicated in increased collagen turnover is ‘secreted protein acidic and rich in cysteine’ (SPARC) since it has been found in high levels in periodontitis patients who present with lesser amounts of bone destruction [20,21,22]. However, most of these correlations are inconsistent as some researchers report increased levels of salivary OC, ON, and CTX [9,10] in periodontal disease, while others report otherwise [11].

In Saudi Arabia, the use of cigarette smoking was known to be around 12.2% in 2005 among adults [23] while it increased to 21.4% in 2018 [24]. Cigarette smoking is a major risk factor for periodontal disease and is serious health concern in this Kingdom. Smoking exerts its deleterious effects by various mechanisms such as increased oxidative stress and reduced antioxidant defenses, heightened inflammatory activity, and reduced host defense mechanisms that impair reparative capacities of tissues. Oxidative stress involves dysfunction in the production and manifestation of reactive oxygen species (ROS) along with the repair of the resulting damage [25]. These may be due to the deleterious effects resulting from production of peroxides and free radicals that damage proteins, lipids, and DNA. Although our research group recently reported the applicability of salivary biomarkers for bone turnover for detecting patients with alveolar bone loss in diabetic patients [7], there are no such data available regarding the feasibility of salivary biomarkers in detecting periodontal disease in smokers. The rational for this study was the fact that biomarkers for bone turnover vary in health and disease. Consequently, the aim of the study was to determine the feasibility of using salivary biomarkers for bone turnover to discriminate healthy patients from periodontitis. The null hypothesis was that there is no significant difference between biomarkers for bone turnover in health and disease. Hence, the objective of this study was to determine the levels and diagnostic accuracy of salivary OC, ON, and CTX as potential markers of periodontal bone destruction among current adult smokers to understand the potential role of these biomarkers in periodontal disease. Correlation and diagnostic accuracy of these biomarkers with bleeding on probing (BOP) and probing pocket depth (PPD) were also evaluated.

## 2. Materials and Methods

### 2.1. Study Design and Sampling Strategy

In this cross-sectional study, 276 consecutive male and female patients who attended the Outpatient Department (OPD) at the College of Dentistry, King Khalid University (April to October 2019) were evaluated for eligibility for participation. Generalized periodontitis (GP) patients (according to the new classification by the American Academy of Periodontology [26]) between the age group of 25 to 44 years who had no relevant medical history were considered for further examination. Those who agreed to the protocol signed the informed consent and were included in the study as shown in Figure 1.

The Institutional Review Board approved the protocol, and the Scientific Research Committee at the College of Dentistry, King Khalid University, Abha, Saudi Arabia gave the ethical approval (approval no. SRC/ETH/2018-19/106). The study was conducted in full accordance with ethical principles, including the World Medical Association Declaration of Helsinki (version 2008). The selection criteria were based on the latest World Workshop on the Classification of Periodontal and Peri-implant Diseases and Conditions [27]. The patients were grouped as healthy control (periodontally healthy and non-smokers; *n* = 30 in group I), Non-smoker periodontitis (generalized periodontitis (GP) stage I–III with non-smokers; *n* = 30 in group II) and smoker periodontitis (current cigarette smokers [27] with GP stage I–III; *n* = 30 in group III). Group I participants were bystanders/volunteers who had healthy, intact periodontium and were non-smokers. Those with bleeding on probing (BOP) ≤10%, no clinical attachment loss (CAL), bone loss (BL) ≤3 mm, and no history of smoking at all were included in this study. Group II consisted of patients with BOP ≥10% and BL >3 mm in ≥ 30% teeth and did not give any history of smoking at all. Group III consisted of patients with BOP ≥10% and BL >3 mm ≥30% teeth and were current cigarette smokers (current smokers using ≥10 cigarettes/day, at the time of the study) [27]. Patients with any systemic diseases or medications, periodontal therapy within three months, systemic antibiotics or nonsteroidal anti-inflammatory drugs within the last three months, current use of corticosteroids, all categories of smoking other than current smoker including former smokers and lactating, or pregnant ladies were excluded from the study. Patients needing complex rehabilitation (stage IV) according to the new classification of periodontitis and less than 20 remaining teeth were also excluded due to the vast extent of periodontal destruction [12]. Furthermore, patients with reduced periodontium in non-periodontitis cases and successfully treated stable periodontitis patients, third molars, mal-aligned teeth, teeth with ill-defined cementoenamel junction (CEJ) due to reasons including caries or restorations, and those with overhang/subgingival restorations were also excluded from this study. All smokers included in the study were given definitive treatment and were later referred to smoking cessation clinics.

### 2.2. Collection of Salivary Samples For Estimation of Bone Biomarkers

Whole saliva (unstimulated) was collected by the drool method from every patient in the morning (9 to 10 am) before clinical examination into a sterile vial (5 mL) using a small funnel to prevent spillage. Patients were asked to rinse the mouth thoroughly with distilled water for 30 s and expectorate, after which saliva was collected by the passive drool method. The participants were asked to be relaxed and to avoid eating, drinking, chewing gum, and brushing their teeth 1 h before the procedure to prevent any external factors that could affect the amount of saliva secretion [28]. The collected sample would be excluded if the quantity of saliva was insufficient or if the patient ate or drank 1 h before saliva collection. The vials with salivary samples were sealed, labeled for identification, and placed in a Styrofoam box containing ice after which it was sent to the Clinical Biochemistry Laboratory at the College of Medicine, King Khalid University, for storage (at −80 °C) until further analysis by an experienced senior member in clinical biochemistry. Antibody sandwich ELISA (Enzyme-Linked Immunosorbent Assay) was performed for analyzing the presence of OC, ON, and CTX in the collected salivary samples using ELISA kits. ELISA kits (commercially available) for human ICTP (Cross-Linked C-telopeptide of Type I Collagen) (catalog no: E-EL-H0835) and human OC/BGP (osteocalcin) ELISA Kit (catalog no: E-EL-H1343) were obtained from Elabscience Biotechnology Inc, Houston, TX, USA, while human SPARC (osteonectin) ELISA^®^ Kit (Catalog No: AB220654) was obtained from Abcam, Cambridge, UK, for estimation.

Estimation of the biochemical constituents was done by adding 100 µL of respective standard working solution to one of the wells in the designated micro ELISA plates pre-coated with an antibody for each of the three biochemical constituents. Samples (100 µL) were added to all the other wells, and the plate was covered with the sealer provided in the kit and incubated for 90 min at 37 °C. The liquid was removed out of each well without any washing. Immediately, 100 μL of biotinylated detection working solution (containing antibody specific to human ICTP, osteocalcin, or osteonectin) was added to each well in the respective plates designated for each of these constituents. Plates were again covered with the plate sealer, mixed gently, and incubated for 1 h at 37 °C. The solution from each well was decanted, and 350 µL of wash buffer was added to each well, soaked for 1~2 min, and the solution was decanted from each well and patted it dry against clean absorbent paper. This wash step was repeated 3 times. Horse radish peroxidase (HRP) conjugate working solution (100 µL) was added to each well, and the plate was covered with sealer and incubated for 30 min at 37 °C. This solution was again decanted from each well, and the wash process was repeated five times as before. Then, 90 μL of substrate reagent was added to each well, covered again with a new plate sealer, and incubated for about 15 min at 37 °C, protecting the plates from light. Finally, 50 μL of stop solution was added to each well. The optical density (OD value) of each well was read at once with a micro-plate reader (ELISA reader) set to 450 nm. Standard curves were drawn for each of the constituents separately with different dilutions of standard solutions, and the concentration of that constituent in the saliva was calculated from this standard graph. Manufacturer instructions were followed to estimate the levels of these constituents, and the analytical performance of the assay(s) was validated to confirm the manufacturer’s analytical performance claims. The minimum detection limit of CTX was 0.0312 ng/mL, osteocalcin was 1.25 ng/mL, and osteonectin was 0.0125 ng/mL.

### 2.3. Periodontal Clinical Examination

Full-mouth clinical and radiographic parameters were recorded independently by two experienced periodontists (blinded to smoking status) for each patient included in the study. BOP was assessed based on the gingival bleeding index given by Ainamo and Bay, and the results were expressed in percentages. PPD, recession, and CAL were determined with the help of a periodontal probe (University of Michigan O probe with William’s markings) using a double-pass technique. While calculating the CAL, cases involving gingival recession due to trauma; caries involving the CEJ; the presence of CAL on the distal aspect of a second molar or due to malposition or extraction of a third molar; a draining endodontic lesion involving the marginal periodontium; and vertical root fracture were excluded. The intraclass correlation coefficient value for the inter-observer reliability was 0.90 (excellent agreement) in terms of both the degree of correlation and agreement between measurements. Since radiographic assessment of BL was the primary dependable variable of this study, cases with generalized bone destruction (BL more than 3 mm in 30% or more teeth [29,30]) were considered for groups II and III. BL was measured from the cementoenamel junction (CEJ) to the crest of alveolar bone in the worst affected tooth for each patient.

Assessment of BL was done independently by two different periodontists who were blinded to the study groups. The procedure was standardized by parallel technique (Kodak Ultraspeed Dental Film, Eastman Kodak, Rochester, NY, USA) with a Siemens Heliodent MD model X1744 (Sirona Dental Systems, GmbH D-64625, Bensheim, Germany), and the X-ray machine was used at 70 kV and 7 mA. Images of digital periapical radiographs were analyzed at 5× magnification using a software program (Adobe Photoshop™ software (version 7.0, Adobe Systems Inc., San Jose, CA, USA). CEJ on the tooth and crest of alveolar bone were used as a reference point for measuring BL. The intraclass correlation coefficient value for the inter-observer reliability was 0.96, which reflects an excellent agreement in terms of both the degree of correlation and agreement between measurements. BL was measured in millimeters based on the worst affected tooth. The arithmetic means of measurements for each patient by two different examiners were considered. A score of 0 was given when the distance between CEJ and alveolar crest of bone was within 3 mm (since physiological bone levels are known to range from 1.0 to 3.0 mm apical to the cementoenamel junction [23]), while a score of 1 was given to cases that had more than 3 mm bone loss when measured from CEJ.

### 2.4. Statistical Analysis

To summarize the data, results were expressed as mean ± SD (quantitative data), and as numbers and percentages (qualitative data). A normality assumption was made. Chi-square test was done to compare qualitative variables and quantitative variables between the study groups. Student’s *t*-test was used to compare mean values among the two groups. Pearson’s correlation analysis was done to assess the correlation between periodontal variables with salivary biomarkers of bone turnover, and the results were expressed in terms of *p*-value and Pearson’s coefficient (r). The receiver operator characteristic (ROC) curve was fitted into the data to evaluate the predictability of biomarkers and was based on the binary assumption. Two models (sets for data) were used for ROC: a) normal cut-off for defining health (BOP: 10% BL: 15%; PPD: 4 mm) and b) higher cut-off for health (BL: 33.33%; PPD: 6 mm). The discriminative performance of these two models was based on the capability to differentiate between subjects with and without periodontitis, among healthy controls, smoker periodontitis, and non-smokers periodontitis. The area under the receiver operating characteristic (ROC) curve (AUC) was used to express the accuracy, which was obtained by plotting sensitivity against 1-specificity. A model that perfectly discriminates between healthy and diseased has an AUC of 1.0, while 0.5 indicates discrimination by chance alone. Youden Index (YI) has been described along with AUC too, as YI describes the maximum potential effectiveness of a biomarker [31].

## 3. Results

Out of the total 276 consecutive patients examined, 122 were found eligible to participate. Fifteen patients refused a radiographic assessment, and 10 did not cooperate fully in saliva collection, thus there was a total of 25 dropouts. Finally, 90 patients (30 in each group) completed the study between April to October 2019. Mean ages in groups I, II, and III were 32.70 ± 7.09, 34.93 ± 7.31, and 35.13 ± 7.40 respectively. Saliva collection was done before recording the clinical parameters on the same day. The clinicians who measured radiographic bone loss and statistician were blinded to the diagnosis of the patients in all three groups.

Descriptive statistics showed that, statistically, there was no significant difference between study groups in terms of socio-economic status, age and oral hygiene practices, *p* > 0.05 (Table 1). Among the three groups, the majority of participants were toothbrush users, practiced brushing once daily, and took around 1–2 min for the same. Table 2 shows that all periodontal variables (BOP, PPD, and BL) had a statistically significant difference (*p* < 0.01). Salivary OC, ON, and CTX increased from group I to group II but showed a decrease in group III. Among these, salivary OC showed the least percentage of variation. Following this, a correlational analysis was performed between periodontal variables with salivary biomarkers (OC, ON, and CTX) as shown in Table 3. Salivary parameters showed only a weak to moderate positive correlation with periodontal parameters (i.e., OC, ON, and CTX) with BOP (r = 0.60,0.46 and 0.48 respectively), PPD (r = 0.40,0.32, and 0.36 respectively), and BL (r = 0.58,0.38, and 0.51 respectively), which were statistically significant (*p* < 0.01).

OC, ON, and CTX discriminated total periodontitis (periodontitis in non-smokers and smokers combined) from healthy controls with excellent sensitivity (80%–91.67%) and good specificity (76.67%–86.67%) based on the different cut-off values obtained from coordinates of ROC (Table 4a). Similarly, periodontitis among smokers was discriminated from healthy controls with good sensitivity (60.0%–90.0%) and specificity (79.31%–100.0%). Excellent discrimination was found between healthy and total periodontitis as determined by the AUC of ROC. It ranged between 0.833–0.873 in terms of OC (AUC: 0.871; 95% CI: 0.783–0.932), ON (AUC: 0.833; 95% CI: 0.739–0.903), and CTX (AUC: 0.873; 95% CI: 0.787–0.934). ROC curve gave excellent discrimination between healthy from smoker periodontitis in terms of OC (AUC: 0.870; 95% CI: 0.757–0.943), ON (AUC: 0.754; 95% CI: 0.625-–0.857), and CTX (AUC: 0.844; 95% CI: 0.726–0.925) as shown in Table 4b.

Among the three biomarkers, CTX at 20.19 ng/mL gave the best overall discrimination with a sensitivity, specificity, and YI of 91.67%, 86.67%, and 0.783, respectively, in terms of healthy against total periodontitis (*p* = 0.001), while OC at 15.25 ng/mL gave a very similar AUC (0.871) but a slightly lower YI (0.667). Likewise, among smoker periodontitis too, this same level of CTX (20.19 ng/mL) gave excellent discrimination in terms of AUC = 0.844 (95% CI: 0.726–0.925), sensitivity (83.33%), specificity (89.66%), and YI (0.729). Marginally higher AUC (0.870) was obtained in this category with 15.25 ng/mL of OC (*p* < 0.0001).

However, when higher cut-offs were used for BL (33.33%) and PPD (6 mm), salivary OC gave the best discrimination in terms of both total periodontitis (periodontitis in non-smokers and smokers combined) and smoker periodontitis as against healthy controls (Table 5a–c). With a cut-off of BL at 33.33%, OC at a concentration of 16.45 ng/mL showed the highest discriminatory values (AUC: 0.708; 95% CI: 0.603–0.799; Se: 94.12; Sp: 46.58, and YI: 0.407 between healthy and total periodontitis. A slightly higher AUC of 0.809 (95% CI: 0.686–0.900; Se: 80.0%, Sp: 73.47%, and YI: 0.534) was noticed when 9.24 ng/mL of salivary OC was considered as the cut-off among smoker periodontitis. Like the results of BL (33.33%), a cut-off of 6 mm PPD also gave excellent discrimination (AUC: 0.811; 95% CI: 0.688–0.901; Se: 92.31%, Sp: 65.22%, and YI: 0.575) with OC (16.45 ng/mL) among healthy and smoker periodontitis.

Figure 2, Figure 3, Figure 4 and Figure 5 show the ROC curve for the periodontal parameters, while Table 4a,b and Table 5a–c show the sensitivity, specificity, area under the curve (AUC) at 95% CI, and YI for these periodontal parameters.

## 4. Discussion

Our primary finding is that salivary OC at 15.25 ng/mL gave the best discrimination between healthy controls and smoker periodontitis. It was also found that salivary OC, ON, and CTX correlated positively with the periodontal parameters (BOP, PPD, and BL). Similar correlation studies have been done earlier [9,10,11,12,13,14,15], but no one has reported the diagnostic accuracy of these biomarkers in young adult smokers with periodontitis. In this present study, smokers showed more severe periodontal parameters as compared to non-smoker periodontitis. Although similar clinical results have been found in smokers as compared with nonsmokers [32], few studies showed that smokers had higher plaque accumulation and fewer sites with BOP than non-smokers [33,34]. Various reasons for increased periodontal destruction in smokers such as an increase in the number of certain bacteria when the pH in the saliva is decreased, alterations in salivary function, and varied taste have been found in participants with reduced salivary flow [35]. Interaction of cigarette smoking with the oxidative status of the body and changes in the antioxidant levels influence the initiation and progression of periodontal disease [36].

In the current study, salivary CTX levels were decreased in smokers with periodontitis as compared to non-smokers. This is contradictory to the reports of other studies [10,33] in which no changes were found in the ICTP levels based on the smoking status. Variations in the salivary levels might also be due to the difference in the method of the assay (radioimmunoassay or ELISA). CTX is specific to bone resorption as pyridinoline cross-links represent a potentially valuable diagnostic marker in bone remodeling. The reasonable diagnostic accuracy of CTX levels in oral fluids makes it as a potential marker of increased bone loss [13,14]. There is evidence that CTX correlates with clinical parameters of periodontal disease as well as reduces following periodontal therapy, thus indicating its role in ongoing tissue breakdown [37,38,39].

In the present study, lower levels of salivary OC were found in smoker periodontitis patients than non-smoker periodontitis. This finding is in line with the previous reports that revealed lower salivary OC levels in smoker periodontitis than non-smoker counterparts [32,33]. However, higher levels of salivary OC were found in the present study among smoker periodontitis patients than healthy controls. This could be because OC is a specific marker of bone formation that recruits osteoclasts to sites of newly formed bone and thus may function as a negative regulator. Suppression of salivary OC levels by smoking partly is due to the detrimental effects of smoking on periodontal health. Higher levels of salivary OC in smoker periodontitis of this present study correlates with our previous study [12]. Components of tobacco smoke (nicotine and non-nicotine) have been shown to downregulate osteoblast activity [40]. Lower levels of serum OC levels found in a study (radioimmunological method based on an antibody to human osteocalcin) [40] among smokers than in non-smokers are similar to our study, suggesting that smoking may induce osteoblast depression by a mechanism that either is due to hormonal changes or direct effects. This may also be partly explained based on the unfavorable impact of smoking on mRNA expression of OC [41]. Decreased levels of salivary OC were seen associated with increasing years of cigarette smoking [33]. This could be one mechanism that explains the negative effect of smoking on the healing of alveolar bone, which worsens with increasing years of cigarette smoking [42].

We chose 15.25 ng/mL (OC) as the “best” cut-off for discriminating smoker periodontitis from healthy controls in this study, as we aimed to eventually create a test that maximized sensitivity (90.0%). However, upon calculation of YI, 20.19 ng/mL of CTX (AUC = 0.844; 95% CI: 0.726–0.925; Se: 83.33%, Sp: 89.66%) had the best YI (0.729), since YI measures the overall diagnostic accuracy of a test. This is the maximum vertical distance or difference between the ROC curve and the diagonal and is a function of the sensitivity and specificity chance line [31]. CTX at 20.19 ng/mL gave excellent discrimination between healthy and total periodontitis. Higher cut-off values for BL (33.33%) and PPD (6 mm) also gave excellent discrimination in the case of salivary OC, both in terms of total periodontitis (periodontitis in non-smokers and smokers combined) and smoker periodontitis against healthy controls.

ON levels in this study increased from healthy controls to non-smoker periodontitis in a way that corresponded to the increase in alveolar bone loss and PPD. This is contrary to a few prior studies were a negative correlation of SPARC/osteonectin was observed in terms of bone loss in periodontitis patients [9,38]. However, these results are similar to our previous study that included periodontitis patients with diabetes [12]. The potential for SPARC/osteonectin to facilitate alveolar bone healing through the deposition of collagen could explain the increased levels of ON in the diseased group. It is found that lack of SPARC/osteonectin causes a reduction in total collagen, and its production is known to be affected by periodontal disease [15].

Although some limitations must be attributed to the present study, the results are still novel and encouraging. A thorough search of related studies revealed that this study is the first of its kind from this region and among the early few globally that evaluates the feasibility of detecting these salivary biomarkers among smokers. Based on the coordinates obtained from the ROC curve, the best cut-off points have been suggested here, and we have shown for the first time that salivary bone biomarkers are linked with alveolar bone loss in smokers. The geographic variations of this southern region of Saudi Arabia, Asir (situated at an elevation of 2270 meters above sea level) present with physiological challenges such as fall in barometric pressure with an increase in altitude, the body’s adaptive mechanism to long term hypoxia, and resultant acclimatization. However, there are no earlier studies of salivary biomarkers in periodontitis patients from any other Arab nation. The results of this study can be used as a reference range standard for salivary OC, ON, and CTX for screening smokers for periodontitis. The sample size, although small, provides a meaningful statistical evaluation of the data [43]. It should be emphasized that although clinical and radiographic evaluation detects tissue destruction of the past, they remain the mainstay for the diagnosis of periodontal disease. This could be because several risk factors or confounders might influence the biochemical marker level such as neural regulation of salivary secretions, degree of dehydration, and biological rhythm. Hence, to date, salivary diagnostics need to be supported by clinical attachment levels and radiographic bone loss.

The convenience sampling method that was used in this had its advantages; it may have had some undesirable impact on the external validity of the results. Only young adults were included in this study. This is because they substantially contribute to the main fraction of the smoking population in Saudi Arabia and also to optimize the control of confounding factors that may influence periodontal destruction associated with age differences [44]. Data from smoking cessation clinics in this region [45] prompted us to take up exclusive cigarette smokers for this study as they formed the majority among smokers (61.6%). Hence, those patients who smoked Tinbak and Shishah, as well as former smokers, were not included in the present study.

In this sense, newer studies are needed to elucidate the link between salivary bone biomarkers and alveolar bone levels in various groups with larger sample sizes, preferably former and passive smokers. These studies also should be done utilizing samples from various times on the same day to understand the diurnal variation of these salivary components and during seasons of fasting such as the holy month of Ramadan. The knowledge acquired from this study can potentially lead to the development of point-of-care chairside tests and the applicability of salivary biomarkers to determine the risk of bone loss. Further research could pave the way for prediction of susceptibility to alveolar bone loss and screen for periodontal diseases that would require minimal clinical training and resources. This is valuable to the clinician not only in identifying current smokers who are susceptible to periodontitis but also former smokers and patients undergoing smoking cessation. Such innovations can improve oral health services and the monitoring of high-risk patients in large communities, especially underserved communities. Post-menopausal osteoporotic women with periodontitis who are at a high risk of systemic and oral bone loss are also potential candidates for these types of salivary diagnostic studies to have a better understanding of the association of salivary biomarkers to periodontitis. It is recommended that community-based interventions such as tobacco health education, health promotion, and counseling be implemented widely to tackle the menace of tobacco smoking.

In conclusion, we have evaluated the feasibility of using salivary biomarkers for bone turnover to discriminate healthy patients from periodontitis. This study demonstrates that salivary biomarkers of bone turnover (OC, ON, and CTX) have a remarkable difference between periodontally healthy and diseased groups. These correlated positively with PPD and BL with a remarkable difference between healthy and diseased patients. It can be suggested that OC, ON, and CTX have high discriminating power to identify the radiographic bone loss in periodontitis, with 15.25 ng/mL of salivary OC offering the best discrimination between healthy controls and smoker periodontitis. Since there is a need to detect periodontal disease at an early stage, the reference range presented here may be used as a screening tool for identifying high-risk patients with initial stage of periodontitis. This, in turn, would enhance the feasibility of early detection of periodontal disease as it is crucial in the clinical management of periodontal patients.

## Figures and Tables

**Figure 1 biomolecules-10-00380-f001:**
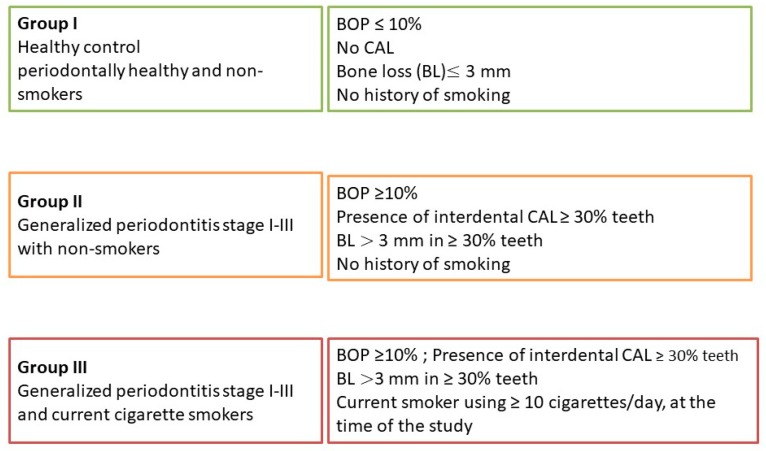
Flow chart of patient recruitment.

**Figure 2 biomolecules-10-00380-f002:**
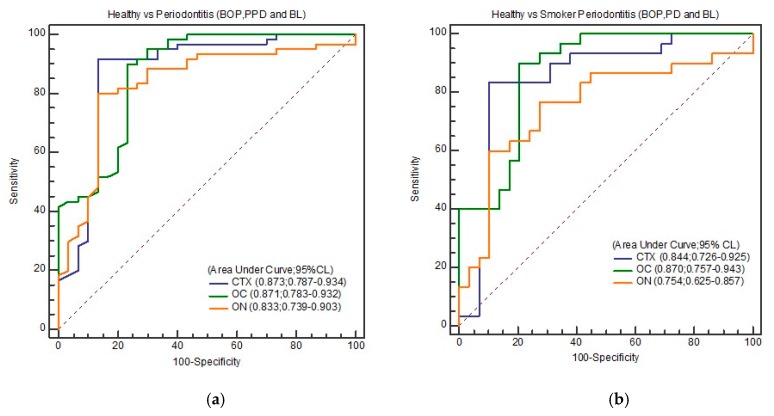
(**a**) ROC curve of healthy and total periodontitis (BOP, PD, and BL); (**b**) ROC curve of healthy and smoker periodontitis (BOP, PD, and BL).

**Figure 3 biomolecules-10-00380-f003:**
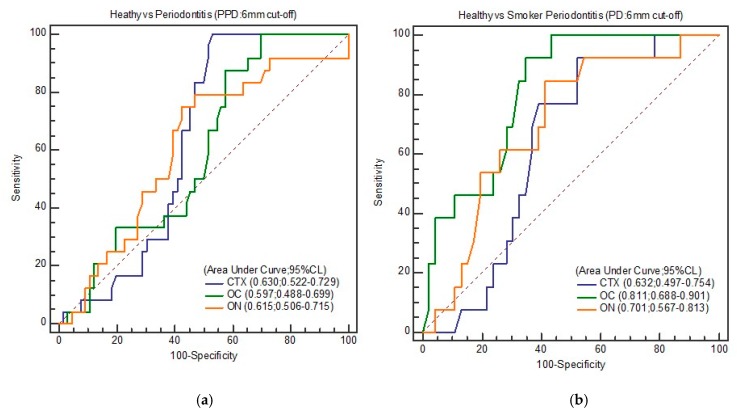
(**a**) ROC curve of healthy and total periodontitis (PD: 6 mm); (**b**) ROC curve of healthy and smoker periodontitis (PD: 6 mm).

**Figure 4 biomolecules-10-00380-f004:**
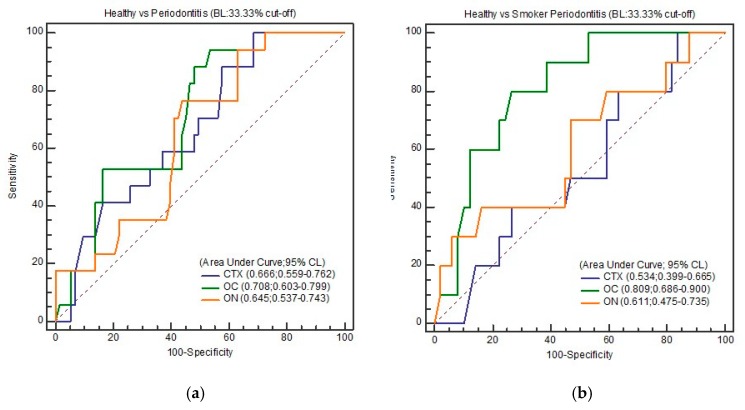
(**a**) ROC curve of healthy and total periodontitis (BL: 33.33%); (**b**) ROC curve of healthy and smoker periodontitis (BL: 33.33%).

**Figure 5 biomolecules-10-00380-f005:**
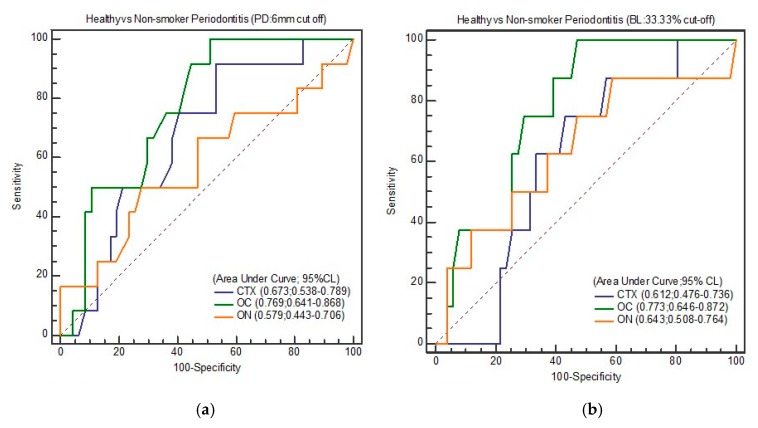
(**a**) ROC curve of healthy and non-smoker periodontitis (PD: 6 mm); (**b**) ROC curve of healthy and non-smoker periodontitis (BL: 33.33%).

**Table 1 biomolecules-10-00380-t001:** Descriptive analysis of demographic variables between study groups.

Variables		Group I (*n* = 30)	Group II (*n* = 30)	Group III (*n* = 30)	Total	*p*-Value
**Gender**	Male	18 (60.0)	14 (46.7)	22 (73.3)	54 (60.0)	0.11 ^ns^
	Female	12 (40.0)	16 (53.3)	8 (26.7)	36 (40.0)	
**Age**	Mean ± SD	32.70 ± 7.09	34.93 ± 7.31	35.13 ± 7.40	-	0.35 ^ns^
**Smoking Status**	Absent	30	30	0	60 (66.7)	0.000 **
	Present	0	0	30	30 (33.3)	
**Teeth Present**	Mean ± SD	28.93 ± 2.33	29.67±1.93	29.13 ± 2.08	-	0.38 ^ns^
**Oral Hygiene Tool**	No oral hygiene	7 (23.3)	8 (26.7)	12 (40.0)	27 (30.0)	0.429 ^ns^
	Toothbrush	15 (50.0)	16 (53.3)	15 (50.0)	46 (51.1)	
	Miswak twig	8 (26.7)	6 (20.0)	3 (10.0)	17 (18.9)	
**Frequency of Brushing**	Once daily	16 (53.3)	16 (53.3)	13 (43.3)	45 (50.0)	0.298 ^ns^
	Twice daily	10 (33.3)	7 (23.3)	6 (20.0)	23 (25.6)	
	Infrequent	4 (13.3)	7 (23.3)	11 (36.7)	22 (24.4)	
**Time Taken for Brushing**	0–1 min	1 (3.3)	0	2 (6.7)	3 (3.3)	0.637 ^ns^
	1–2 min	18 (60.0)	23 (76.7)	20 (66.7)	61 (67.8)	
	2–5 min	11 (36.7)	7 (23.3)	8 (26.6)	27 (28.9)	
**BOP**	<10%	30 (100.0)	0	0	30 (33.33)	0.000 **
	≥10%	0	30 (100.0)	30 (100.0)	60 (66.7)	
**PPD**	<4.0 mm	30 (100.0)	0	0	30 (33.33)	0.000 **
	≥4.0 mm	0	30 (100.0)	30 (100.0)	60 (66.7)	
**Bone Loss**	<15%	30 (100.0)	0	0	30 (33.33)	0.000 **
	≥15%	0	30 (100.0)	30 (100.0)	60 (66.7)	

**Note**: ns: not significant; * *p* < 0.05: significant; ** *p* < 0.001: highly significant; SD: standard deviation.

**Table 2 biomolecules-10-00380-t002:** Comparative analysis of periodontal variables and salivary biomarkers of bone turnover between study groups.

Variable	Periodontal Variables (Mean ± SD)			*p* Value
	Group I	Group II	Group III	
**BOP**	7.20 ± 2.26	56.99 ± 18.44	64.02 ± 15.70	0.000 ^¶^
**PPD (mm)**	2.53 ± 0.62	5.2 ± 0.67	5.3 ± 0.84	0.000 ^¶^
**Bone Loss (mm)**	1.2 ± 0.61	3.9 ± 0.59	4.17 ± 0.68	0.000 ^¶^
**Bone Loss (%)**	9.65 ± 4.51	29.33 ± 6.9	31.69 ± 7.25	0.000 ^¶^
**Salivary Biomarkers of Bone Turnover (Mean ± SD)**				
	**Group I**	**Group II**	**Group III**	
**Osteocalcin (ng/mL)**	10.97 ± 5.80	24.99 ± 8.97	23.11 ± 8.63	0.000 ^¶^
**Osteonectin (ng/mL)**	59.47 ± 21.26	109 ± 20.48	82.53 ± 31.17	0.000 ^¶^
**CTX (ng/mL)**	21.08 ± 17.75	61.90 ± 11.57	39.21 ± 10.28	0.000 ^¶^

**Note:** * *p* value < 0.05; ^€^
*p* value < 0.01; ^¶^
*p* value < 0.001; PPD: periodontal pocket depth; SD: standard deviation.

**Table 3 biomolecules-10-00380-t003:** Correlational analysis of periodontal variables with salivary biomarkers of bone turnover.

Variable	BOP	PPD	Bone Loss	CTX	Osteocalcin	Osteonectin
**Osteocalcin**	0.000 ^¶^ (0.60)	0.001 ^€^ (0.40)	0.000 ^¶^ (0.58)	0.000 ^¶^ (0.51)	-	0.000 ^¶^ (0.50)
**Osteonectin**	0.010 * (0.46)	0.012 * (0.32)	0.000 ^€^ (0.38)	0.000 ^€^ (0.48)	0.000 ^¶^ (0.50)	-
**CTX**	0.001 ^€^ (0.48)	0.004 ^€^ (0.36)	0.000 ^¶^ (0.51)	-	0.000 ^¶^ (0.51)	0.000 ^€^ (0.48)

**Note**: * *p* value < 0.05; ^€^
*p* value < 0.01; ^¶^
*p* value < 0.001; *p*-value (Pearson coefficient); BOP: bleeding on probing; PPD: probing pocket depth.

**Table 4a biomolecules-10-00380-t004a:** Cut off values, area under curve of ROC, and Youden Index of CTX, OC, and ON in discriminating healthy from total periodontitis based on BL, BOP, and PPD.

	Cut-off (ng/mL)		Se (%)	Sp (%)	AUC	95% CI	P	YI
**Healthy vs. Total Periodontitis** **(BOP: 10% BL: 15%; PPD: 4 mm)**	**CTX**	18.3	91.67	70.00	0.873	0.787–0.934	0.001	0.783
		20.19	91.67	86.67				
		22.14	88.33	86.67				
	**OC**	9.21	100	56.67	0.871	0.783–0.932	0.001	0.667
		15.25	90.00	76.67				
		25.99	41.67	100.0				
	**ON**	55.0	93.33	53.33	0.833	0.739–0.903	0.001	0.667
		68.2	80.00	86.67				
		98.2	45.0	90.0				

Se: sensitivity; Sp: specificity; ROC: receiver operator characteristic; AUC: area under curve; 95% CI: 95% confidence interval (CI); YI: Youden Index = Se + Sp − 1; PPD: probing pocket depth; BOP: bleeding on probing; BL: bone loss; CTX: deoxypyridinoline-containing degradation fragment of the C-terminal telopeptide region of type I collagen; OC: osteocalcin; ON: osteonectin.

**Table 4b biomolecules-10-00380-t004b:** Cut off values, area under curve of ROC, and Youden Index of CTX, OC, and ON in discriminating healthy from smoker periodontitis based on BL, BOP, and PPD.

	Cut-off (ng/mL)		Se (%)	Sp (%)	AUC	95% CI	P	YI
**Healthy vs. Periodontitis** **(BOP: 10% BL: 15%; PPD: 4 mm)**	**CTX**	17.37	90.0	68.97	0.844	0.726–0.925	<0.0001	0.729
		20.19	83.33	89.66				
		22.14	76.67	89.66				
	**OC**	9.21	100	58.62	0.870	0.757–0.943	<0.0001	0.693
		15.25	90.00	79.31				
		25.99	41.67	100.0				
	**ON**	55.0	83.33	58.62	0.754	0.625–0.857	<0.0001	0.496
		68.2	60.00	89.66				
		98.2	45.0	90.0				

Se: sensitivity; Sp: specificity; ROC: receiver operator characteristic; AUC: area under curve; 95% CI: 95% confidence interval (CI); YI: Youden Index = Se + Sp − 1; PPD: probing pocket depth; BOP: bleeding on probing; BL: bone loss; CTX: deoxypyridinoline-containing degradation fragment of the C-terminal telopeptide region of type I collagen; OC: osteocalcin; ON: osteonectin.

**Table 5a biomolecules-10-00380-t005a:** Cut off values, area under curve of ROC, and Youden Index of CTX, OC, and ON in discriminating healthy from total periodontitis based on BL (33%) and PPD (6 mm).

	Cut-off (ng/mL)		Se (%)	Sp (%)	AUC	95% CI	P	YI
**BL (33.33%)**	**CTX**	17.37	100	31.51	0.666	0.559–0.762	0.013	0.315
		42.36	64.71	52.05				
		62.14	41.18	83.56				
	**OC**	16.45	94.12	46.58	0.708	0.603–0.799	0.001	0.407
		18.59	70.59	54.79				
		28.22	52.94	83.56				
	**ON**	88.64	76.47	56.16	0.645	0.537–0.743	0.032	0.326
		92.14	64.71	58.90				
		138.7	17.65	100				
	**Cut-off (ng/mL)**		**Se (%)**	**Sp (%)**	**AUC**	**95% CI**	**P**	**YI**
**PPD (6 mm)**	**CTX**	20.19	100	46.97	0.630	0.522–0.729	0.023	0.469
		30.11	83.33	50.0				
		39.54	75.0	54.55				
	**OC**	9.45	100	30.30	0.597	0.488–0.699	0.117	0.303
		15.64	87.50	42.42				
		16.47	70.83	45.45				
	**ON**	77.5	79.17	53.03	0.615	0.537–0.643	0.084	0.327
		88.2	75.00	57.58				
		89.14	66.67	59.09				

Se: sensitivity; Sp: specificity; ROC: receiver operator characteristic; AUC: area under curve; 95% CI: 95% confidence interval (CI); YI: Youden Index = Se + Sp − 1; PPD: probing pocket depth; BOP: bleeding on probing; BL: bone loss; CTX: deoxypyridinoline-containing degradation fragment of the C-terminal telopeptide region of type I collagen; OC: osteocalcin; ON: osteonectin.

**Table 5b biomolecules-10-00380-t005b:** Cut off values, area under curve of ROC, and Youden index of CTX, OC, and ON in discriminating healthy from non-smoker periodontitis based on BL (33%) and PPD (6 mm).

	Cut-off (ng/mL)		Se (%)	Sp (%)	AUC	95% CI	P	YI
**BL (33.33%)**	**CTX**	17.37	87.50	43.14	0.612	0.476–0.736	0.211	0.319
		20.19	75.00	56.86				
		71.56	0	100				
	**OC**	15.64	100	52.94	0.773	0.646–0.872	<0.0001	0.529
		16.19	87.50	54.90				
		29.21	37.50	92.16				
	**ON**	55.6	87.50	41.18	0.643	0.508–0.764	0.231	0.287
		56.33	75.0	45.10				
		124.6	25.0	96.08				
	**Cut-off (ng/mL)**		**Se (%)**	**Sp (%)**	**AUC**	**95% CI**	**P**	**YI**
**PPD (6 mm)**	**CTX**	17.37	91.67	46.81	0.673	0.538–0.789	0.030	0.384
		18.15	75.00	46.81				
		45.21	50.0	78.72				
	**OC**	11.41	100	48.94	0.769	0.641–0.868	<0.0001	0.489
		15.64	91.67	55.32				
		26.12	50	89.32				
	**ON**	58.11	66.67	53.19	0.579	0.443–0.706	0.450	0.223
		88.64	50.00	72.34				
		133.28	16.67	100				

Se: sensitivity; Sp: specificity; ROC: receiver operator characteristic; AUC: area under curve; 95% CI: 95% confidence interval (CI); YI: Youden Index = Se + Sp − 1; PPD: probing pocket depth; BOP: bleeding on probing; BL: bone loss; CTX: deoxypyridinoline-containing degradation fragment of the C-terminal telopeptide region of type I collagen; OC: osteocalcin; ON: osteonectin.

**Table 5c biomolecules-10-00380-t005c:** Cut off values, area under curve of ROC, and Youden index of CTX, OC, and ON in discriminating healthy from smoker periodontitis based on BL (33%) and PPD (6 mm).

	Cut-off (ng/mL)		Se (%)	Sp (%)	AUC	95% CI	P	YI
**BL (33.33%)**	**CTX**	15.88	80.0	36.78	0.534	0.399–0.665	0.729	0.167
		17.37	70.0	40.82				
		20.19	50.0	53.06				
	**OC**	11.41	100	46.94	0.809	0.686–0.900	<0.001	0.534
		19.24	80.0	73.47				
		25.99	60.0	87.76				
	**ON**	55.6	80.0	40.82	0.611	0.475–0.735	0.298	0.238
		58.11	70.0	53.06				
		108.36	30.0	93.88				
	**Cut-off (ng/mL)**		**Se (%)**	**Sp (%)**	**AUC**	**95% CI**	**P**	**YI**
**PPD (6 mm)**	**CTX**	17.37	92.31	47.31	0.632	0.497–0.754	0.0732	0.401
		20.19	76.92	60.87				
		71.56	100	0				
	**OC**	16.45	92.31	65.22	0.811	0.688–0.901	<0.001	0.575
		18.23	84.62	67.39				
		29.21	38.46	95.65				
	**ON**	57.65	84.62	54.35	0.701	0.567–0.813	0.009	0.433
		58.11	84.62	58.70				
		92.14	53.85	80.43				

Se: sensitivity; Sp: specificity; ROC: receiver operator characteristic; AUC: area under curve; 95% CI: 95% confidence interval (CI); YI: Youden Index = Se + Sp − 1; PPD: probing pocket depth; BOP: bleeding on probing; BL: bone loss; CTX: deoxypyridinoline-containing degradation fragment of the C-terminal telopeptide region of type I collagen; OC: osteocalcin; ON: osteonectin.

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
