# Peer review of "Salivary Osteocalcin as Potential Diagnostic Marker of Periodontal Bone Destruction among Smokers"

_biomolecules, 2020, doi:10.3390/biom10030380_

Round 1
Reviewer 1 Report
In the manuscript entitled: “Salivary Osteocalcin as potential diagnostic markers of periodontal bone destruction among smokers” the authors assessed the levels and diagnostic accuracy of salivary Osteocalcin (OC), Osteonectin (ON), and deoxypyridinoline-containing degradation fragment of the C-terminal telopeptide region of type I collagen (CTX) in adult smokers with periodontal bone destruction.
In their study, systemically healthy ninety patients (groups I: healthy, II: periodontitis with non- smokers and III: periodontitis with current smokers) were enrolled.
The authors found a positive correlation for OC, ON, and CTX with PPD and BL. Smoker periodontitis were best discriminated from healthy controls using 15.25 ng/ml of OC. 16.45 ng/ml of OC gave excellent discrimination with among healthy and smoker periodontitis when PD at 6mm was considered as cut- off.
The authors concluded that the best discrimination between healthy controls and smoker periodontitis was obtained at 15.25 ng/ml of salivary OC.
Major comments:
In general, the idea and innovation of this review, regards the analysis of osteocalcin as potential diagnostic markers of periodontal bone destruction is interesting, because the role of salivary biomarkers is quite validated but further studies on this topic could be an innovative issue in this field could be open an innovative matter of debate in literature by adding new information. Moreover, there are few reports in the literature that studied this interesting topic with this kind of study design.
The study was well conducted by the authors; However, there are some concerns to revise that are described below.
The introduction section resumes the existing knowledge regarding the main salivary biomarkers of periodontitis.
However, as the importance of the topic, the reviewer strongly recommends to update the literature through read, discuss and cites in the references with great attention all of those recent interesting articles, that helps the authors to better introduce and discuss the aim of the study in light of also of the role of the delivery agents that could influence the oral dysbiosis progression: 1) Isola G, Polizzi A, Alibrandi A, Indelicato F, Ferlito S. Analysis of Endothelin-1 Concentrations in Individuals with Periodontitis. Sci Rep. 2020 Feb 3;10(1):1652. doi: 10.1038/s41598-020-58585-4. 2) Isola G, Polizzi A, Muraglie S, Leonardi R, Lo Giudice A. Assessment of Vitamin C and Antioxidant Profiles in Saliva and Serum in Patients with Periodontitis and Ischemic Heart Disease. Nutrients. 2019 Dec 4;11(12). pii: E2956. doi: 10.3390/nu11122956
The authors should be better specified, at the end of the background section, the rational of the study and the aim of the study with the null hypothesis.
In the materials and methods, the authors should better define the exclusion criteria of the analysed sample and the radiographic assessment of BL.
The discussion section appears well organized with the relevant paper that support the conclusions, even if the authors should better discuss the importance of the salivary biomarkers during periodontitis. The conclusion should reinforce in light of the discussions.
In conclusion, I am sure that the authors are fine clinicians who achieve very nice results with their adopted protocol. However, this study, in my view does not in its current form satisfy a very high scientific requirement for publication in this journal and requests some revisions before a further re-evaluation of the manuscript.
Minor Comments:
Abstract:
- Better formulate the conclusion section by adding the background
Introduction:
- Please refer to major comments;
Discussion
- Please add a specific sentence that clarifies the results obtained in the first part of the discussion
- Lines 377-388: Please reorganize this paragraph that is not clear
Author Response
Reply to reviewers
We would like to thank the Honorable Editor and reviewers for taking the time to reviewer our manuscript. We have done our best to make it acceptable. Hope it is satisfactory to you.
Reviewer-1
- The introduction section resumes the existing knowledge regarding the main salivary biomarkers of periodontitis. However, as the importance of the topic, the reviewer strongly recommends to update the literature through read, discuss and cites in the references with great attention all of those recent interesting articles, that helps the authors to better introduce and discuss the aim of the study in light of also of the role of the delivery agents that could influence the oral dysbiosis progression:
Authors’ response: Thank you for your constructive comments. The following references have been added in the introduction part based on your suggestions.
1) Isola G, Polizzi A, Alibrandi A, Indelicato F, Ferlito S. Analysis of Endothelin-1 Concentrations in Individuals with Periodontitis. Sci Rep. 2020 Feb 3;10(1):1652. doi: 10.1038/s41598-020-58585-4.
2) Isola G, Polizzi A, Muraglie S, Leonardi R, Lo Giudice A. Assessment of Vitamin C and Antioxidant Profiles in Saliva and Serum in Patients with Periodontitis and Ischemic Heart Disease. Nutrients. 2019 Dec 4;11(12). pii: E2956. doi:10.3390/nu11122956
- The authors should be better specified, at the end of the background section, the rational of the study and the aim of the study with the null hypothesis.
Authors’ response: The rational for this study was the fact that biomarkers for bone turnover vary in health and disease. Consequently, the aim of the study was to determine the feasibility of using salivary biomarkers for bone turnover to discriminate healthy patients from periodontitis and the null hypothesis was that there is no significant difference between biomarkers for bone turnovers in health and disease.
- In the materials and methods, the authors should better define the exclusion criteria of the analysed sample and the radiographic assessment of BL.
Authors’ response: Exclusion criteria for patients have been explained on page 4 (line number 118-128) and for the analyzed sample on page 4 (lines 138-139) while the exclusion criteria for radiographic assessment of BL is explained on page 5 (lines 176-179).
- The discussion section appears well organized with the relevant paper that support the conclusions, even if the authors should better discuss the importance of the salivary biomarkers during periodontitis. The conclusion should reinforce in light of the discussions.
Authors’ response: Thank you for your encouraging comments. Please note that we have modified the conclusion as follows-
“In conclusion, we have evaluated the feasibility of using salivary biomarkers for bone turnover to discriminate healthy patients from periodontitis. This study demonstrates that salivary biomarkers of bone turnover (OC, ON, and CTX) have a remarkable difference between periodontally healthy and diseased groups. These correlated positively with PPD and BL with a remarkable difference between healthy and diseased patients. It can be suggested that OC, ON, and CTX have high discriminating power to identify the radiographic bone loss in periodontitis with 15.25 ng/ml of salivary OC offering the best discrimination between healthy controls and smoker periodontitis. Since there is a need to detect periodontal disease at an early stage, the reference range presented here may be used as a screening tool for identifying high-risk patients with initial stage of periodontitis. This in turn would enhance the feasibility of early detection of periodontal disease as it is crucial in the clinical management of periodontal patients.”
- In conclusion, I am sure that the authors are fine clinicians who achieve very nice results with their adopted protocol. However, this study, in my view does not in its current form satisfy a very high scientific requirement for publication in this journal and requests some revisions before a further re-evaluation of the manuscript.
Authors’ response: We sincerely hope that we have been able to address to all your queries satisfactorily. We once again thank you for taking the time to review this paper and for giving us your valuable comments.
Minor Comments:
Abstract:
- Better formulate the conclusion section by adding the background
Authors’ response: In conclusion, we have evaluated the feasibility of using salivary biomarkers for bone turnover to discriminate healthy patients from periodontitis. This study demonstrates that salivary biomarkers of bone turnover (OC, ON, and CTX) have a remarkable difference between periodontally healthy and diseased groups. These correlated positively with PPD and BL with a remarkable difference between healthy and diseased patients. It can be suggested that OC, ON, and CTX have high discriminating power to identify the radiographic bone loss in periodontitis with 15.25 ng/ml of salivary OC offering the best discrimination between healthy controls and smoker periodontitis. Since there is a need to detect periodontal disease at an early stage, the reference range presented here may be used as a screening tool for identifying high-risk patients with initial stage of periodontitis. This in turn would enhance the feasibility of early detection of periodontal disease as it is crucial in the clinical management of periodontal patients.
- Introduction:
Please refer to major comments-
Authors’ response: Kindly find the changes done in the introduction section based on major comments.
- Discussion
Please add a specific sentence that clarifies the results obtained in the first part of the discussion
Authors’ response: With due respect, we would like to state that we have made all efforts to minimize direct use of results in the discussion. Instead, we have used inferences based on our results.
- Lines 377-388: Please reorganize this paragraph that is not clear
Authors’ response: The convenience sampling method that was used in this study would have had its advantages; it may have had some undesirable impact on the external validity of the results. Only young adults were included in this study, this was because they substantially contribute to the main fraction of the smoking population in Saudi Arabia and to optimize the control of confounding factors that may influence periodontal destruction associated with age differences.
Thank you once again.

Reviewer 2 Report
The authors present a study on the potential application of Osteocalcin a salivary diagnostic marker among smokers and non-smokers with periodontitis. The study is well conducted. However, the authors should pay attention to the following:
- The manuscript is very extensive. The authors should reduce the manuscript to a more concise manner to make it easier for a broad range of readers
- English language needs revision by a native speaker (example the following the sentence: The arithmetic means of measurements for each patient by two different examiners)
- The authors should mention specifically what was the original number of participants in each group and how many drop-outs were from each group. The authors mention this information in a very general manner. Also, this reviewer finds it a bit strange that after drop-outs the numbers in each group completely matched (i.e. 30 participants in each group)
Author Response
Reply to reviewers
We would like to thank the Honorable Editor and reviewers for taking the time to reviewer our manuscript. We have done our best to make it acceptable. Hope it is satisfactory to you.
Reviewer-2
- The manuscript is very extensive. The authors should reduce the manuscript to a more concise manner to make it easier for a broad range of readers
Authors’ response: Thank you for your comment. We agree that the manuscript in extensive, but this is because we did not want to miss out on any aspect of the study. Once the manuscript is accepted, kindly suggest us where all to delete so that it can be made concise.
- English language needs revision by a native speaker (example the following the sentence: The arithmetic means of measurements for each patient by two different examiners)
Authors’ response: Please note that English language revision has been done.
- The authors should mention specifically what was the original number of participants in each group and how many drop-outs were from each group. The authors mention this information in a very general manner. Also, this reviewer finds it a bit strange that after drop-outs the numbers in each group completely matched (i.e. 30 participants in each group)
Authors’ response: Please note that, of the total 276 consecutive patients examined, only 122 were found eligible to participate. 15 patients refused a radiographic assessment and 10 did not cooperate fully in saliva collection, thus there total 25 dropouts. That left us with 276-154= 122-25=97. Since we wanted equal number of participants in each group, we finally recruited 30 patients each in every group.
Thank you once again.

Reviewer 3 Report
The topic of this article entitled “Salivary Osteocalcin as potential diagnostic markers of periodontal bone destruction among smokers” is interesting. Nevertheless, this reviewer would suggest some improvements, before further considerations.
- The Authors stated: “Various cells and biochemical molecules including osteoblasts, receptor activator of nuclear factor kappa B ligand, osteoprotegerin, osteoclasts, bone morphogenetic proteins, resorption products of organic skeletal matrix and inorganic skeletal matrix markers are involved.” Please, discuss better something about the regenerative strategies with stem cells in bone remodeling/repairing (Please, see and discuss: Tatullo, M.; Codispoti, B.; Pacifici, A.; Palmieri, F.; Marrelli, M.; Pacifici, L.; Paduano, F. Potential use of human periapical cyst-mesenchymal stem cells (hpcy-mscs) as a novel stem cell source for regenerative medicine applications. Front Cell Dev Biol 2017, 5, 103.) and about novel strategies to replace bone tissue (Please, see and discuss: Kerativitayanan, P.; Tatullo, M.; Khariton, M.; Joshi, P.; Perniconi, B.; Gaharwar, A.K. Nanoengineered Osteoinductive and Elastomeric Scaffolds for Bone Tissue Engineering. ACS Biomaterials Science & Engineering 2017 3 (4), 590-600) taking into consideration the novel approaches with nanotechnologies (please see and discuss: Barry M, Pearce H, Cross L, Tatullo M, Gaharwar AK. Advances in Nanotechnology for the Treatment of Osteoporosis. Curr Osteoporos Rep. 2016;14(3):87–94.)
- The authors stated “Smoking exerts its deleterious effects by various mechanisms such as increased oxidative stress and reduced antioxidant defences, heightened inflammatory activity, reduced host defense mechanism impaired reparative capacities of tissues.” They should describe more in detail the systemic conditions related to oxidative stress impairing (Please, see and discuss: Tatullo M, et al. Relationship between oxidative stress and "burning mouth syndrome" in female patients: a scientific hypothesis. Eur Rev Med Pharmacol Sci. 2012 Sep;16(9):1218-21.)
In the methods: authors should report in a clear manner the inclusion criteria.
English must be improved.
Conclusions must be improved and something should be added about the translational applications of the results described in this research.
Author Response
Reply to reviewers
We would like to thank the Honorable Editor and reviewers for taking the time to reviewer our manuscript. We have done our best to make it acceptable. We hope it is satisfactory to you.
Reviewer 3
- The authors stated: “Various cells and biochemical molecules including osteoblasts, receptor activator of nuclear factor kappa B ligand, osteoprotegerin, osteoclasts, bone morphogenetic proteins, resorption products of organic skeletal matrix and inorganic skeletal matrix markers are involved.” Please, discuss better something about the regenerative strategies with stem cells in bone remodeling/repairing (Please, see and discuss: Tatullo, M.; Codispoti, B.; Pacifici, A.; Palmieri, F.; Marrelli, M.; Pacifici, L.; Paduano, F. Potential use of human periapical cyst-mesenchymal stem cells (hpcy-mscs) as a novel stem cell source for regenerative medicine applications. Front Cell Dev Biol 2017, 5, 103.) and about novel strategies to replace bone tissue (Please, see and discuss: Kerativitayanan, P.; Tatullo, M.; Khariton, M.; Joshi, P.; Perniconi, B.; Gaharwar, A.K. Nanoengineered Osteoinductive and Elastomeric Scaffolds for Bone Tissue Engineering. ACS Biomaterials Science & Engineering 2017 3 (4), 590-600) taking into consideration the novel approaches with nanotechnologies (please see and discuss: Barry M, Pearce H, Cross L, Tatullo M, Gaharwar AK. Advances in Nanotechnology for the Treatment of Osteoporosis. Curr Osteoporos Rep. 2016;14(3):87–94.)
Authors’ response: Thank you for your positive comments. Please note that these references have been included. The following sentence has been added on page 2 (lines 53-55)
“Management of periodontal disease includes various strategies such as regenerative techniques using bone grafts, stem cells for bone remodeling/repairing and approaches using nanotechnologies.
- The authors stated “Smoking exerts its deleterious effects by various mechanisms such as increased oxidative stress and reduced antioxidant defences, heightened inflammatory activity, reduced host defense mechanism impaired reparative capacities of tissues.” They should describe more in detail the systemic conditions related to oxidative stress impairing (Please, see and discuss: Tatullo M, et al. Relationship between oxidative stress and "burning mouth syndrome" in female patients: a scientific hypothesis. Eur Rev Med Pharmacol Sci. 2012 Sep;16(9):1218-21.)
Authors’ response: Thank you for your positive comment. Please note that this reference has been included. The following sentence has been added on page 2 (lines 78 to 81).
“Smoking exerts its deleterious effects by various mechanisms such as increased oxidative stress and reduced antioxidant defenses, heightened inflammatory activity, reduced host defense mechanism impaired reparative capacities of tissues. Oxidative stress involves dysfunction in the production and manifestation of reactive oxygen species (ROS) along with the repair of the resulting damage. These may be due to the deleterious effects due to the production of peroxides and free radicals that damage proteins, lipids, and DNA.”
- In the methods: authors should report in a clear manner the inclusion criteria.
Authors’ response: We have tried our best to make the inclusion criteria clear.
- English must be improved.
Authors’ response: Please note that English language revision has been done.
- Conclusions must be improved and something should be added about the translational applications of the results described in this research
Authors’ response: Please note that English language revision has been done.
Authors’ response: Since there is a need to detect periodontal disease at an early stage, Tthe reference range presented here may be used as a screening tool for identifying high-risk patients with initial at an early stage of periodontitis. This, in turn, would enhance the feasibility of early detection of periodontal disease as it is can be crucial in the clinical management of periodontal patients.
Thank you once again.

Round 2
Reviewer 1 Report
In the R1 version of the manuscript entitled: “Salivary Osteocalcin as potential diagnostic markers of periodontal bone destruction among smokers” the authors followed all the issues suggested by the reviewer. Though the changes based on the reviewer comments, almost of the criticisms were carefully analysed and solved.
I have carefully evaluated all parts of the manuscript. I believe that the article, in this version, is now adequate for publication in this journal.
Reviewer 3 Report
Authors have successfully addressed the reviewer's suggestions